# Independent and Interactive Roles of Immunity and Metabolism in Aortic Dissection

**DOI:** 10.3390/ijms242115908

**Published:** 2023-11-02

**Authors:** Siyu Li, Jun Li, Wei Cheng, Wenhui He, Shuang-Shuang Dai

**Affiliations:** 1School of Medicine, Chongqing University, Chongqing 400044, China; 2Department of Biochemistry and Molecular Biology, Third Military Medical University (Army Medical University), Chongqing 400038, China; 3Department of Cardiac Surgery, The First Affiliated Hospital of Third Military Medical University (Army Medical University), Chongqing 400038, China

**Keywords:** aortic dissection, immunity, metabolism

## Abstract

Aortic dissection (AD) is a cardiovascular disease that seriously endangers the lives of patients. The mortality rate of this disease is high, and the incidence is increasing annually, but the pathogenesis of AD is complicated. In recent years, an increasing number of studies have shown that immune cell infiltration in the media and adventitia of the aorta is a novel hallmark of AD. These cells contribute to changes in the immune microenvironment, which can affect their own metabolism and that of parenchymal cells in the aortic wall, which are essential factors that induce degeneration and remodeling of the vascular wall and play important roles in the formation and development of AD. Accordingly, this review focuses on the independent and interactive roles of immunity and metabolism in AD to provide further insights into the pathogenesis, novel ideas for diagnosis and new strategies for treatment or early prevention of AD.

## 1. Introduction

Aortic dissection (AD) is a dangerous cardiovascular disease. The incidence rate in the general population is 3–5 cases/100,000 persons, and the incidence rate in the middle-aged and elderly population can be as high as 10 cases/100,000 persons. The mortality rate of untreated type A acute AD (AAD) patients after symptom onset is 1% to 2% per hour [1,2,3,4]. Without clinical intervention, the mortality rate is as high as 50% within 48 h after onset. The long-term survival of patients who survive an acute aortic event is reduced, with an increased risk of death that extends to 90 days from diagnosis [5,6]. The aortic wall is composed of the intima, media and adventitia from the inside to the outside. The inner membrane is composed of endothelial cells (ECs) and the subendothelial layer. The media is mainly composed of elastic fibers, collagen fibers and vascular smooth muscle cells (VSMCs). The adventitia of the aorta is composed of loose connective tissue, which contains elastic fibers, collagen fibers and fibroblasts. Under physiological conditions, the three aortic membranes stick closely to each other and jointly bear the pressure of blood flow in the blood vessels. However, when there is a break in the inner membrane, the impact of blood can further tear and expand the gap, causing the three-layer membrane to separate and resulting in the formation of AD. The pathogenesis of AD is not completely clear. The accepted view is that, in response to some internal and external factors, an entry forms in the inner membrane, and blood enters the media through the entry, separating the inner and outer membranes. The occurrence and development of this event are mainly caused by vascular inflammation, extravascular matrix degradation and VSMC apoptosis. Hypertension, atherosclerosis, Marfan syndrome and smoking are all related risk factors for AD [7,8,9]. Among them, approximately two-thirds of patients with dissection have high blood pressure, 50% of AD patients under 40 years of age have Marfan syndrome and approximately 20% of patients have thoracic aortic aneurysm (AA) or a family history of AD [10].

Apart from genetics, lifestyle habits and background diseases, it is important to focus on the cellular and molecular mechanisms of AD, and abnormal immunity and metabolism are two essential aspects involved in AD initiation and progression [11,12,13]. Recent clinical and basic studies have shown that extravascular matrix degradation and VSMC apoptosis are exacerbated as the degree of inflammation increases [14,15]. This finding suggests that the inflammatory response may play an important role in the early onset of AD and can activate multiple pathological processes to further promote the onset of AD. Furthermore, immune cells such as macrophages, lymphocytes and neutrophils are often detected in the media and adventitia of AD tissues [16,17]. Moreover, abnormal carbohydrate, lipid and amino acid metabolism has gained increasing attention recently. Immunometabolism has gradually become a new research hotspot in the medical field and has been widely used in the research of diabetes, cancer and cardiovascular disease. Choosing a more reasonable strategy to regulate immune metabolism and maintain natural immune homeostasis is critical. Immunity and metabolism exert modulatory effects independently and interactively. Infiltrated immune cells not only promote the secretion of matrix metalloproteinases (MMPs) and adhesion molecules but also release reactive oxygen species (ROS), leading to changes in the metabolism in the microenvironment and VSMC apoptosis, which ultimately leads to the development of AD [18,19,20]. In contrast, alterations in immune and parenchymal cell metabolism change the immune microenvironment and immune cell function, which are associated with the development of AD [21,22]. Therefore, this review elucidates the pathological mechanism of AD development through a novel perspective of immunity and metabolism to provide a new perspective for comprehensive cognition, diagnosis and treatment of AD.

## 2. Immunity in AD

### 2.1. Innate Immunity and Associated Signaling in AD

#### 2.1.1. Neutrophils

Neutrophils have phagocytic and chemotactic functions and play a role in the inflammatory cascade after AAD. Lauren et al. showed that neutrophil infiltration into the adventitia occurred within 12 h after AD and peaked from 12 to 24 h [23]. Moreover, clinical studies have shown that the neutrophil-to-lymphocyte ratio (NLR) can be used as a prognostic predictor of AD [24,25]. Apart from this, neutrophils are the main cells that secrete MMPs. Some researchers have shown that the accumulation of MMP-9 leads to disorders of ECM metabolism, promoting inflammation and the degradation of elastic fibers and accelerating the expansion and rupture of the dissection [26]. Moreover, neutrophil-derived IL-6 is a potent enhancer of post-AAD adventitial inflammation that leads to increased CRP levels following aortic rupture [27]. Additionally, activated neutrophils can release neutrophil extracellular traps (NETs), serine proteases, histone proteases and ROS intermediates that can promote the progression of vascular inflammation and AD [28,29]. Yang et al. measured the expression levels of three NET markers in AAD with different severities and confirmed that NET levels are a reliable factor in the diagnosis of AAD and are related to the severity and prognosis of the disease [30]. Laridan et al. also reached the same conclusion in a mouse model, in which the presence of NETs could be detected in the early stage of abdominal aortic aneurysm (AAA). Treatment of induced mice with cloamidine, a NET inhibitor, significantly reduced AAA formation [31]. The new study demonstrates that neutrophil elastase (NE) promotes thoracic aortic dissection (TAD) development and aortic dissection by enhancing inflammatory cell recruitment/migration into the vessel wall, increasing aortic inflammation and promoting SMC phenotypic switching. Importantly, this study also highlights the therapeutic value of the NE-TBL1x signaling axis in patients with high-risk TAD [32]. Research has shown that depleting neutrophils with antibodies or reducing neutrophil infiltration into vascular tissue in mice could reduce MMP production and significantly reduce the incidence of AD [33].

#### 2.1.2. Monocytes

Monocytes are a heterogeneous cell population that can be categorized into three subpopulations with different phenotypic and functional properties: “classical” (CD14^++^CD16^−^), “intermediate” (CD14^++^CD16^+^) and “nonclassical” (CD14^+^CD16^+^). Furthermore, classical monocytes are significantly increased, whereas intermediate monocytes are significantly decreased in AAD [34]. Factors such as oxidative stress, cytokines, viral or bacterial infections, high blood sugar or high low-density lipoprotein (LDL) levels can activate the vascular endothelium, induce monocyte recruitment, promote the secretion of the chemokines CCL2, CCL5 and CCL7 and upregulate adhesion molecules such as intercellular cell adhesion molecule-1 (ICAM-1) and vascular cell adhesion molecule-1 (VCAM-1), which can promote the rolling of circulating blood mononuclear cells, adhesion to ECs and migration into the tissue [35,36]. Monocytes that migrate from the blood to the tissues play a phagocytic role, produce inflammatory factors and then differentiate into tissue macrophages and dendritic cells [37,38]. It has been reported that IL-6 can induce monocytes to differentiate into macrophages and can mediate Th17 production through the IL-6 signal transducer and activator of transcription 3 (STAT3) pathway [39]. Activated monocytes and platelet receptor glycoprotein Ib alpha (GPIbα) participate in local thrombin amplification through coagulation factor XI (FXI), thereby promoting the development of vascular inflammation and hypertension and accelerating the progression of AD [40]. Additionally, low-density lipoprotein receptor-related protein 8 (LRP8) is released from monocytes and can induce vascular inflammation and endothelial dysfunction, leading to AD [41].

#### 2.1.3. Macrophages

Pathophysiological processes, including macrophage migration and infiltration into the aorta, differentiation or phenotype transformation, the secretion of various inflammatory factors and the release of extracellular matrix (ECM)-degrading proteins, are closely related to the occurrence and development of AD [42]. Inflammatory factors in the peripheral blood and vascular wall of patients with Stanford type A dissection were examined by Del Porto et al., and the results showed that macrophages were the main inflammatory cells that infiltrated aortic tissues, the ECM was degraded, and the immune response played an important role in this process [43]. Consistently, Darrell et al. showed that the infiltration of macrophages in AAD tissue was significantly higher than that in chronic AD tissue [44]. M1 (proinflammatory) and M2 (anti-inflammatory) macrophages play different roles in the pathological process of AD [45]. Cytotoxic M1 macrophages release a variety of proinflammatory factors, ROS and nitric oxide synthase (NOS) to further tissue damage; however, M2 macrophages exert anti-inflammatory effects by inhibiting T cell proliferation and the phagocytosis of apoptotic neutrophils and reducing the production of proinflammatory factors [46]. Recent studies have shown that IL-6 activates STAT3 through negative regulation of cytokine signal transduction inhibitor 3 and changes the gene expression programs of macrophages and VSMCs, thereby regulating the inflammatory response and affecting AD [47]. In contrast, IL-5 overexpression in macrophages attenuates the inflammatory response and SMC apoptosis, thereby inhibiting the development of AD [48]. In addition, researchers found that DNA from damaged SMCs was phagocytosed by macrophages and activated the STING pathway via interferon regulatory factor 3 to induce the expression of MMP-9, which promoted the degradation of the ECM to accelerate AD [49]. Additionally, adenosine deaminase acting on RNA (ADAR1) is involved in macrophage activation and AAA formation through degradation of Drosha protein [50].

#### 2.1.4. Mast Cells

Mast cells promote the development and progression of AD through the release of inflammatory factors. Gao et al. found that mast cells could secrete the proinflammatory factors interferon-γ (IFN-γ) and MMPs, which cause VSMC apoptosis and vascular remodeling, leading to the formation of AA [51]. In addition, Springer et al. showed that in vivo mast cells could partially and transiently regulate systemic IL-6 homeostasis [52]. Swedenborg et al. also proved that mast cells play an important role in AA. Studies indicate that mast cells are the major source of cathepsin G in AA and promote the activation of the renin–angiotensin system through cathepsin G and chymase, which contribute to VSMC apoptosis and the release of proteolytic enzymes, thereby accelerating AA [53].

### 2.2. Adaptive Immunity and Associated Signaling in AD

#### 2.2.1. T Cells/Th Cells

T cells/Th cells have been shown to induce VSMC apoptosis and MMP synthesis [54], which suggests that these cells play a role in the development of AD [55,56]. Clinical studies have shown high levels of CD3^+^, CD4^+^, CD8^+^ and CD45^+^ T cells in the aortic tissue of AD patients, indicating that T cell activation is involved in the development of AD [57]. CD4^+^ T cells can differentiate into different subpopulations under different conditions, such as T helper 1 (Th1), Th2, Th17 and T regulatory (Treg) cells [58]. In addition, the numbers of Th1, Th9, Th17 and Th22 cells and the expression of their transcription factors were elevated in AD patients, while the levels of Th2 and Treg cells and their transcription factors were decreased [56]. This finding suggested that Th2 cells and Treg cells may protect against the development of AD. Several clinical studies have shown that Treg cells play a critical role in inflammation associated with arterial injury in a variety of cardiovascular diseases, such as AAD and symptomatic carotid artery stenosis [59,60]. Studies have confirmed that Treg cells can play both anti-inflammatory and proinflammatory roles, which may be attributed to the heterogeneity of Treg cells. For instance, IL-10-producing Treg cells may protect against aortic wall rupture through anti-inflammatory effects on AAD, whereas CD25^+^ Treg cells are increased and mediate inflammation in patients with symptomatic carotid stenosis [61,62]. Furthermore, it was confirmed that Th17 and Treg cells share common precursor cells, and both cells require transforming growth factor-β (TGF-β) signaling for their initial differentiation, but they perform opposite functions, in which IL-6 plays an essential role [63]. IL-6 and IL-17 produced by Th17 cells are significantly elevated in AD tissue, and IL-6 can promote the transformation of Th0 cells into Th17 cells [64]. Therefore, inhibiting IL-6 to reduce Th0 cells to Th17 cells has a positive impact on controlling AAD onset and progression [65].

#### 2.2.2. B Cells

B cells play a key role in adaptive immunity and participate in the pathogenesis of AD, mainly through their ability to detect and process antigens [66,67]. Saraff et al. revealed that in an angiotensin II (Ang II)-induced AD model, B cells mainly infiltrated the media of the vessel wall. During AD development, B cells are stimulated by antigens to proliferate and differentiate into plasma cells, which synthesize and secrete antibodies and circulate in the blood to perform humoral immune functions. However, the specific role of B cells in AD is still unclear. Schaheen et al. discovered that a lack of mature B cells protects against the formation of AAAs induced by elastase [68]. However, Akshaya et al. showed that there was no significant difference in the incidence of AAs between mice lacking mature B cells and wild-type mice [69]. At present, the role of B cells in AA is still controversial, and more research on their function in AD or AA is needed.

### 2.3. Interactions of Immune Cells in AD

Local inflammatory responses and systemic inflammatory responses in the vessel wall are triggered after AD occurs; this process involves interactions among immune cells. The high expression of chemokines and granulocyte colony-stimulating factor in AD tissue after local inflammation leads to neutrophil infiltration and accelerates the occurrence of lesions in the aortic media [70]. NETs facilitate the secretion of inflammatory factors by macrophages and indirectly activate Th17 cells to promote inflammation [71]. Furthermore, among inflammatory factors, IL-6 can induce the differentiation of monocytes into macrophages and mediate the production of Th17 through the IL-6–STAT3 pathway; Th17 cells can promote the chemotaxis, adhesion and migration of monocytes by mediating inflammatory responses [72]; IL-8 has a role in regulating neutrophil mobilization and migration [73]; IL-1β can promote the secretion of MMPs by immune cells through the phosphorylation of p38 [74]; and exosomes from monocytes are transformed into active macrophages through the NF-κB signaling pathway which continue to secrete a variety of inflammatory substances, proteolytic enzymes and ECM-degrading proteins and continue to damage blood vessels [75]. In addition, immune cells cause changes in the microenvironment, such as the accumulation of homocysteine (Hcy). Liu et al. demonstrated that Hcy stimulates adventitial fibroblasts to produce IL-6 and monocyte chemoattractant protein-1 (MCP-1), thereby recruiting more THP-1 cells to infiltrate the vessel wall [76]. Shao et al. also proved that Hcy induces the secretion of anti-beta 2 glycoprotein I (anti-β2GPI) antibodies by B cells through the toll-like receptor 4 (TLR-4) pathway and then promotes the accumulation of IgG in AAA tissue. In addition, Hcy can polarize inflammatory macrophages and upregulate MMP-2 and MMP-9 expression to promote inflammatory reactions and matrix degradation [77]. In summary, inflammatory cells play an important role in the progression of AD. The direct and indirect interactions between various subsets of immune cells provide a complex inflammatory microenvironment for the initiation and development of AD.

## 3. Metabolisms in AD

### 3.1. Carbohydrate Metabolism

Glucose is an essential source of energy for metabolism. Under normal oxygen conditions, cells prefer glycolysis to the oxidative phosphorylation (OXPHOX) process, known as “Warburg effect”. Lactic acid accumulation and OXPHOX damage caused by the Warburg effect are common in parenchymal cells in AD [78,79]. The inflammatory tissue of AD is often hypoxic [80,81]. Under this condition, the expression of hypoxia-inducible factor (HIF), especially O_2_-sensitive HIF-1α, is enhanced, which plays an important role in the Warburg effect [82,83,84]. Migrating VSMCs have increased glucose transporter 1 (GLUT1) and hexokinase 2 (HK2) expression to promote glycolysis [85,86]. Lactate dehydrogenase (LDHA) is also upregulated, stimulating VSMC proliferation, migration and synthetic phenotypic transformation. In vivo, the inhibitor of LDH and lactate reduced the degradation of elastic fibers, collagen deposition and MMP2/9 production and inhibited the phenotypic transformation of VSMC so that the progression of AD was delayed [87,88]. The NR1D1–ACO2 axis interferes with aberrant tricarboxylic acid (TCA) cycle functions. α-KG, the substrate of ACO2, supplementation is regarded as an effective therapeutic approach for AD which can increase mitochondrial ETC complex expression to prevent AAA formation [89]. Hypoxic conditions cause a shift in glucose flux towards the pentose phosphate pathway (PPP). High levels of NADPH generated via the PPP as part of an antioxidant mechanism protect VSMCs from apoptosis [90]. Fan et al. indicated that a downregulation of TCA cycle metabolites may contribute to lung function damage in AAD [91]. Based on these findings, there is potential for using metabolic therapies to reduce mortality and improve prognosis in AAD in the future. Any targeted intervention that inhibits the Warburg effect or the generation of metabolic intermediates will be a new direction in the treatment of AD.

### 3.2. Lipid Metabolism

It is becoming increasingly clear that high lipid levels predispose individuals to the development of clinical cardiovascular diseases. When the β oxidation of fatty acids (FAs) is suppressed, less acetyl coenzyme A (acetyl-CoA) enters the TCA, thereby inhibiting OXPHOX [92]. Increased concentrations of non-esterified FAs induce endoplasmic reticulum stress, mitochondrial dysfunction, inflammation and cytokine release, which are causative factors of AD [93]. Observational studies found obvious dyslipidemia in AD patients, and the in-hospital mortality of AD patients was correlated with dyslipidemia [94]. The secretion of arachidonic acid (ARA) and increased synthesis of prostaglandin E2 (PGE2) are associated with endothelial dysfunction and vascular inflammation in patients with MFS [95,96]. Likewise, oleic acid (OA) produces the same effects by activating inducible NOS (iNOS), leading to AA production [97]. In contrast, nitro-oleic acid (nitro-OA) inhibits the ERK1/2 and Smad2 pathways and NF-κB overactivation, which significantly reduces aortic dilatation in AD [98]. Oxidized low-density lipoprotein (ox-LDL) activates caspase 1, leading to the activation of IL-1β, and mature IL-1β is released by human VSMCs, thereby triggering vascular disease [99]. Treatment with ferroptosis driven by the excessive lipid peroxidation accumulation can decrease morbidity in AD models [100]. Considering the different roles of various lipid types in promoting health and reducing disease risk, there is growing interest in lipid metabolism. Lipids can not only be diagnostic markers of AD but also target molecules for AD therapy. Statins are considered the most efficient drugs for the treatment of hyperlipidemia. Steinmetz et al. found that statins were beneficial for the risk of AA expansion after using them to treat an AA animal model [101]. Therefore, it was hypothesized that certain lipid interventions could prevent or slow the progression of AD.

### 3.3. Amino Acid Metabolism

Aberrant and dysregulated signaling associated with amino acid metabolism is closely implicated in AD. Wang et al. showed that plasma aminograms were significantly altered in patients with AD [102]. According to epidemiological studies, hyperhomocysteinemia (HHcy) is a hallmark of vascular injury. HHcy stimulates the secretion of IL-6 and MCP-1 and the subsequent recruitment of monocytes/macrophages and promotes adventitial fibroblasts’ transformation into myofibroblasts [103]. Vitamin B restored TGF-β signaling and promoted lysyl-oxidase-mediated collagen maturation in aortic media. Vitamin B can serve as a synergistic drug for the treatment of MFS [104]. S-adenosyl-L-homocysteine cysteine (SAH) is a more sensitive biomarker of cardiovascular disease than Hcy. High levels of SAH in plasma contribute to endothelial dysfunction [105]. Overexpression of solute carrier family 1 member 5 (SLC1A5), a key glutamine (GLN) transporter, results in mTORC1 activation and VSMC proliferation; these changes are attenuated after additional treatments with GLN metabolism inhibitor BPTES [106,107]. Gallina found that AMPA-type glutamate receptors mediate VSMC transformation into the contractile phenotype [108]. In addition, mutations in the glutamate ammonia ligase (GLUL) gene and high glutamate levels increase the risk of cardiovascular disease [109]. Glutamate is converted to proline during the production of anti-inflammatory cytokines such as IL-10 to promote tissue and inflammatory repair [110]. In addition, clinical studies have shown that elevated plasma succinic acid levels can reliably distinguish AAD patients from patients with other diseases [111]. ECs play a critical role in the pathogenesis of AAD by regulating protein S-sulfhydration, and higher plasma H2S levels are associated with a lower risk of AAD [112]. Considering the findings of the current study, the amino acid profile is expected to provide a novel, noninvasive and objective diagnosis for AD.

### 3.4. Interconnection and Cross-Regulation of Metabolism in AD

The metabolism of carbohydrates, lipids and amino acids is linked through a common intermediate product (acetyl-CoA) and a common metabolic pathway, the TCA cycle, to form a whole system; therefore, when there is a metabolic disorder in any one or more of these factors, there is crosstalk with other pathway disorders. As we mentioned previously, the Warburg effect is not only associated with the glycolytic pathway; “errors” in the glycolytic pathway can bring about “mistakes” in amino acid and lipid metabolism. This series of changes promotes the production of ROS, NADPH and NO, and the production of these substances results in various adverse consequences through various pathways, such as parenchymal cell disruption and dysfunction and the activation of inflammatory factors and inflammasomes. These adverse consequences can eventually promote the occurrence and development of AD.

## 4. Synergism or Antagonism between Immunity and Metabolism in AD

### 4.1. Metabolism Shapes the Immune Microenvironment of AD

Immune cells and parenchymal cells need to constantly regulate their own metabolism to perform their corresponding functions in AD. During this process, the production of both cytokines and metabolites alters the microenvironment, further affecting metabolism; for example, the preferential production of lactate via the Warburg effect promotes neutrophil activity [113]. Nox-derived ROS, which are maintained by NADPH produced via the PPP, can effectively induce the formation of NETs [114,115]. The production of α-KG via GLN catabolism for Jmjd3-dependent epigenetic reprogramming is important for alternative activation of M2 macrophages [116]. Lipoprotein a (Lp(a)) mediates a proinflammatory response through oxidized phospholipid (OxPL), which is recognized as a danger-associated molecular pattern by pattern recognition receptors on innate immune cells and induces monocyte trafficking to induce inflammation in the arterial wall [117]. In addition, the acidic environment produced by aerobic glycolysis accelerates the uptake of lipoproteins by macrophages, and more carbon from gluconeogenesis is bound to FAs and sterols, which contributes to the further accumulation of triglycerides (TGs) and reduces TG catabolism in activated macrophages; a series of microenvironmental changes can enhance the proinflammatory effects of macrophages, exacerbating the progression of AD [118]. There have been reports showing that GLN can increase the expression of the NADPH oxidase components p22 (phox), gp91 (phox) and p47 (phox) and meditate the production of superoxide by neutrophils [119]. Antagonizing GLN uptake is considered a novel approach for the treatment of AD [120,121].

### 4.2. Regulatory Effects of Immune Cells on Metabolic Reprogramming in AD Parenchymal Cells

Different immune cell subpopulations differentiate due to metabolic disorders and, in turn, regulate the metabolic reprogramming of parenchymal cells [122]. When AD is accompanied by inflammation, disturbed carbohydrate, lipid and amino acid metabolism in immune cells results in oxidative stress, mitochondrial stress and endoplasmic reticulum stress, and these series of changes prompt cells to produce ROS, NADPH and NO and activate inflammatory vesicles, accelerating EC, VSMC and other parenchymal cell migration and apoptosis, as well as enhancing ECM degradation [123,124,125,126]. Lian et al. revealed that macrophages induced HIF-1α activation in response to fumarate accumulation, which triggers vascular inflammation, metalloproteinase degradation and elastic plate rupture through increased deintegrin and extracellular mesenchymal protein structural domain 17 (ADAM17) expression in humans and mouse models [127,128]. The researchers also identified an enzyme secreted by inflammatory cells called FLp-PLA2, which can hydrolyze glycerophospholipids to produce bioactive lipids, leading to vascular endothelial dysfunction and exacerbating vascular inflammation, and is clinically regarded as an early warning indicator of AD [129,130]. Additionally, in M1-type macrophages, interruption of the TCA cycle causes the accumulation of the intermediate product succinate, which enhances the inflammatory response and further regulates metabolic reprogramming in parenchymal cells [112]. In addition, ox-LDL can promote the glycolytic capacity of macrophages, thereby promoting cholesterol uptake by VSMCs. Cholesterol can regulate Nox isoforms and redox signaling in VSMCs to provoke vascular disease [131]. In addition, T cells regulate EC function during the inflammatory response by secreting a variety of immunomodulatory molecules and cytokines, such as IL-10 and TGF-β, which have anti-inflammatory effects; IL-4 and IL-13 promote the synthesis of FAs and phospholipids, which protect against complement-induced injury and are potential molecules for AD therapy, and maintain mitochondrial function [132].

## 5. Perspectives

With increasing exploration of research strategies and technologies, elucidating the occurrence and development of AD at the cellular and molecular levels will be conducive to understanding its pathogenesis, diagnosis, treatment and prognosis. To date, several studies focusing on immunity and metabolism have revealed the underlying cellular and molecular mechanisms of AD and mainly included three aspects: How do the heterogeneity and multiple functions of immune cells shape inflammation in AD? How do immune cells interact with parenchymal cells to cause vascular remodeling in AD? How does metabolism at the cell and whole-body level affect the pathological course of AD? Although some features are still unclear, these studies provide a variety of insights (Figure 1 and Graphical Abstract). In the future, to achieve better prevention and treatment of AD, two issues should be addressed: on the one hand, the key molecular networks orchestrating immunity and metabolism must be identified, which will elucidate AD pathogenesis and provide potential treatment targets; on the other hand, specific circulating immune cell subsets or their released factors and specific circulating metabolites will provide early warning or diagnostic biomarkers for AD.

## Figures and Tables

**Figure 1 ijms-24-15908-f001:**
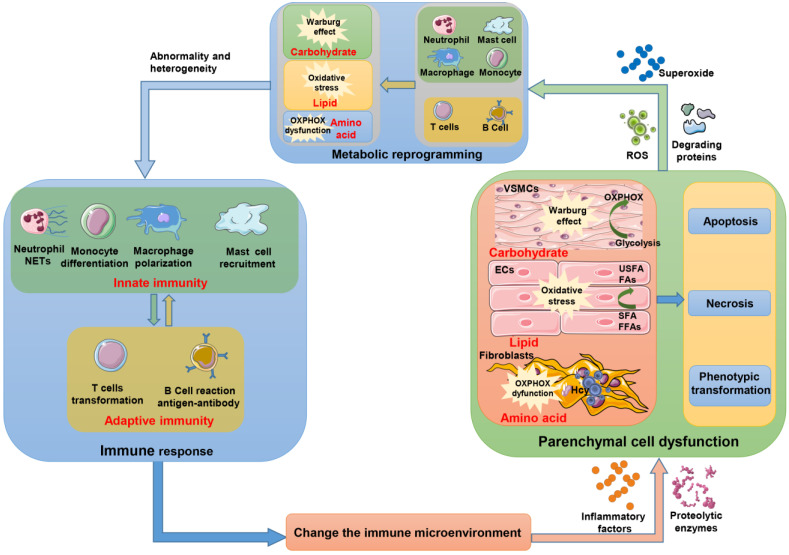
**Schematic diagram of the interaction and influence between the immune metabolic and parenchymal cells.** Innate immunity and adaptive immunity continuously interact to produce an immune response that changes the immune microenvironment; meanwhile, the release of inflammatory factors, proteases, etc., results in abnormal metabolism of parenchymal cells. The glycolysis of glucose in VSMCs becomes the main metabolic pathway rather than OXPHOX, known as the Warburg effect. The excessive accumulation of FFAs and SFA in the lipid metabolism process of ECs can result in oxidative stress. Amino acid products of the TCA cycle are reduced in fibroblasts, whereas HCY accounts for major metabolism which can lead to OXPHPX dysfunction. These metabolic abnormalities eventually lead to apoptosis, necrosis and phenotypic transformation of parenchymal cells. In the process of degrading proteins, ROS and superoxide produced will also cause metabolic reprogramming of immune cells, which will cause abnormality and heterogeneity of immune cells so that they perpetuate inflammation and damage of aortic wall. This vicious cycle will eventually accelerate the process of AD. (NETs: neutrophil extracellular traps, ROS: reactive oxygen species, OXPHOX: oxidative phosphorylation, USFA: unsaturated fatty acid, FAs: fatty acid, SFA: saturated fatty acid, FFAs: free fatty acid, TCA: tricarboxylic acid cycle, Hcy: homocysteine, VSMCs: vascular smooth muscle cells, ECs: endothelial cells).

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
