# Peer review of "Independent and Interactive Roles of Immunity and Metabolism in Aortic Dissection"

_ijms, 2023, doi:10.3390/ijms242115908_

Round 1

Reviewer 1 Report

Comments and Suggestions for Authors

Nice summary of a current knowledge in field of inflammatory response in AD.

However, this paper is only sort of a the scopic review and do not add anything new to the knowledge. Therefore, the chance to be cited in other paper is minimal. I leave the decision about publication to the main editor. Moreover, the presentation of all this information’s in the plain text is somewhat boring to the readers.  

I can recommend a publication only if you consider adding a summarizing diagram presenting influence and interaction between all mentioned in the paper cells, immune and metabolic factors. This might be a difficult task, but finally only this will have a huge and useful impact for this paper. All readers will see the benefit of such a figure. Add this at the end of Perspective chapter – which can support the directions of suggested studies.

Minor issue

Line 31

You mean Type A AD. Do not generalize to type B (be precise)

Line 45

Chang expression “break” to more accurate “entry”

Line 83

Many studies? Is only one listed in references. Change or explain

Line 28

Please explain widely Warburg effect

Reviewer 2 Report

Comments and Suggestions for Authors

The authors focused on the independent and interactive roles of immunity and metabolism in AD to provide further insights into the pathogenesis, novel ideas for diagnosis and new strategies for treatment or early prevention of AD. They reviewed from the 3 points of view: 1) heterogeneity and multiple functions of immune cells. 2)Interactions with parenchymal cells to cause vascular remodeling, 3) Metabolism affecting the pathological course of AD.

General comments

This is a review article addressing “Independent and interactive roles of immunity and metabolism in aortic dissection”.  This is well-written and the conclusions are based on the results. The reviewer has some minor concerns need to be addressed.

1)      Line 99: TAD would need spelling out.

2)      Line 183: “IL-6 can promote the transformation of Th0 cells into Th17 cells”. Then the description that “inhibiting IL-6 to maintain Th17” would be confusing.

3)      Line 240: against.
